# Protein Disulfide Isomerase A3 (PDIA3): A Pharmacological Target in Glioblastoma?

**DOI:** 10.3390/ijms241713279

**Published:** 2023-08-26

**Authors:** Giuliano Paglia, Marco Minacori, Giorgia Meschiari, Sara Fiorini, Silvia Chichiarelli, Margherita Eufemi, Fabio Altieri

**Affiliations:** Department of Biochemical Sciences “A. Rossi Fanelli”, Sapienza University of Rome, P. le Aldo Moro 5, 00185 Rome, Italy; giuliano.paglia@uniroma1.it (G.P.); marco.minacori@uniroma1.it (M.M.); giorgia.meschiari@uniroma1.it (G.M.); sara.fiorini@uniroma1.it (S.F.); silvia.chichiarelli@uniroma1.it (S.C.); margherita.eufemi@uniroma1.it (M.E.)

**Keywords:** protein disulfide isomerase, punicalagin, glioblastoma, PDIA3, ERp57, temozolomide, cancer, inhibitor

## Abstract

The protein disulfide isomerase A3 (PDIA3) is directly or indirectly involved in various physiopathological processes and participates in cancer initiation, progression and chemosensitivity. However, little is known about its involvement in glioblastoma. To obtain specific information, we performed cellular experiments in the T98G and U−87 MG glioblastoma cell lines to evaluate the role of PDIA3. The loss of PDIA3 functions, either through inhibition or silencing, reduced glioblastoma cells spreading by triggering cytotoxic phenomena. PDIA3 inhibition led to a redistribution of PDIA3, resulting in the formation of protein aggregates visualized through immunofluorescence staining. Concurrently, cell cycle progression underwent arrest at the G_1_/S checkpoint. After PDIA3 inhibition, ROS-independent DNA damage and the activation of the repair system occurred, as evidenced by the phosphorylation of H2A.X and the overexpression of the Ku70 protein. We also demonstrated through a clonogenic assay that PDIA3 inhibition could increase the chemosensitivity of T98G and U-87 MG cells to the approved glioblastoma drug temozolomide (TMZ). Overall, PDIA3 inhibition induced cytotoxic effects in the analyzed glioblastoma cell lines. Although further in vivo studies are needed, the results suggested PDIA3 as a novel therapeutic target that could also be included in already approved therapies.

## 1. Introduction

Protein disulfide isomerase A3 (PDIA3), also known as ERp57, is a redox-dependent molecular chaperone involved in the rearrangement of -S-S- bonds in client proteins belonging to the protein disulfide isomerase (PDI) family of enzymes. Although PDIA3 is mainly localized in the endoplasmic reticulum (ER), due to its N-terminal signal peptide, it is ubiquitous within the cell, likely due to a noncanonical ER retention motif (QEDL) at the C- terminal, the presence of a nuclear localization sequence and its capability to interact with a wide range of proteins [1]. The presence of PDIA3 in almost all cellular compartments and the heterogeneity of its substrates concur in its pleiotropic role, which is supported by its direct or indirect involvement in various physio-pathological processes. Recently, the relevance of PDIA3 has been well documented in virus infections [2], neurodegenerative diseases [3], platelet aggregation [4] and oncology [5,6]. It has been reported that PDIA3 participates in cancer initiation, progression and chemosensitivity, and its inhibition has been shown to decrease cell proliferation in several cancers [1,7,8].

Gliomas are brain tumors that originate from glial cells. Glioblastoma, previously known as glioblastoma multiforme (GBM), is the most common. GBM is a highly aggressive tumor of the central nervous system (CNS). The median overall survival of GBM patients is 12–18 months, with only a 4.6% survival rate at 5 years. GBM cells exhibit a rapid proliferation and infiltration, making complete tumor removal through surgical resection challenging [9]. The severity of the disease is graded on a scale of I to IV according to the World Health Organization’s classification of CNS tumors. Grade IV glioma, also known as glioblastoma multiforme (GBM), represents the most malignant form of primary CNS tumor [10]. As with most malignant and rapidly growing tumors, GBM cells are exposed to high levels of cellular stress caused by factors, such as an inadequate blood supply, hypoxia, nutrient deprivation, immune reactions and various therapeutic treatments [11].

Targeting protein disulfide isomerases (PDIs) has emerged as a promising approach for glioblastoma treatment [12,13]. Substantial evidence supports the significant involvement of unfolded protein response (UPR) signaling in GBM oncogenesis and resistance to conventional therapies. The UPR serves as a crucial adaptive mechanism that promotes cell survival in the face of these challenging cellular and environmental conditions [14]. PDIs are considered essential components of UPR signaling, playing a role in regulating cell survival. Consequently, PDIs represent potential therapeutic targets for GBM [15]. Several studies have reported elevated activity of the UPR in GBM, which is sustained by chronic ER stress [12,13,16,17]. The constitutive activation of the ER stress response and the upregulation of ER chaperones have been linked to chemoresistance by abrogating the apoptotic machinery [12,13]. Inhibiting PDI activity can disrupt the formation of disulfide bonds and cause the accumulation of misfolded proteins, triggering cell death mediated by an unresolved UPR [12]. Moreover, the downregulation of DNA repair genes has been reported in response to PDI knockdown, enhancing the efficacy of radiation therapy [12,13]. As demonstrated in numerous papers, PDI inhibition also sensitizes glioblastoma cells to chemotherapy [16,17,18,19].

Regarding the PDIA3 isoform of the PDI family, the expression levels of the protein are highly correlated with glioblastoma. PDIA3 mRNA expression has been shown to be remarkably higher in both low-grade gliomas (oligodendroglioma and astrocytoma) and glioblastomas compared to nontumor controls, as reported in the analysis of gene expression data of gliomas from the Gene Expression Omnibus (GEO) database [20]. Additionally, a qRT-PCR analysis of 99 diffuse glioma specimens (including oligodendrogliomas, oligoastrocytomas, astrocytomas, and glioblastomas) and 11 nontumor specimens highlighted higher PDIA3 expression levels in diffuse glioma specimens, especially in glioblastoma, compared to normal brain tissues [20]. A screen on survival-related genes for glioblastoma conducted on a dataset downloaded from the GEO database revealed that HDAC1 and PDIA3 were highly expressed in GBM tissues [21]. Three independent groups analyzed the survival curves of glioma patients using data from The Cancer Genome Atlas (TCGA) data portal and GEO databases [20,21,22]. In these studies, PDIA3 resulted as a potential key gene affecting the overall survival time of patients with glioblastoma, and high levels of PDIA3 expression have been linked to poor overall survival in glioma patients. We also analyzed the mRNA expression of PDIA3 by using a gene expression profile downloaded from the GEO database (URL accessed on 28/05/2023) and found that PDIA3 was upregulated in glioblastoma compared to normal brain tissue (see Appendix A).

Despite the numerous pieces of evidence showing the correlation between PDIA3 levels and glioblastoma, the molecular mechanisms involving the protein remain obscure. Recently it has been reported that the expression of circRFX3, a novel circular RNA, was substantially increased in glioblastoma cell lines and clinical tissues. This RNA could boost the proliferation, invasion and migration by acting as a competing endogenous RNA for microRNA-587 (miR-587), whose downstream target is PDIA3. The same authors proposed that PDIA3 could regulate the Wnt/β–catenin pathway, which resulted in activation due to PDIA3 overexpression in U-87 MG cells [23].

In the present work, we investigated the role of PDIA3 in glioblastoma and its potential as a therapeutic target for new treatments. The effects of PDIA3 inhibition were analyzed in two glioblastoma cell lines, T98G and U-87 MG cells. These cell lines have commonly been selected as human glioblastoma cell lines in functional studies. U-87 MG and T98G cells display a considerable difference in their reactivity to TMZ, with U-87 MG being highly reactive to TMZ. This difference could be ascribed to the presence in T98G cells of a higher level of O6-methylguanine-DNA methyltransferase (MGMT), an enzyme involved in the demethylation systems that repair TMZ-induced DNA damage [24].

To better understand the role of PDIA3, we used punicalagin (PUN), an active compound of pomegranate, as a PDIA3 inhibitor. In two previous works [25,26] we demonstrated that PUN could specifically bind to PDIA3 and inhibit its disulfide reductase activity. In this study, the efficacy of PUN in a glioblastoma cellular model was tested, and the specificity of PUN on the PDIA3 target was evaluated by performing a PDIA3 silencing before the PUN treatment. The selected glioblastoma cell lines were analyzed before and upon PDIA3 impairment testing of cell viability and proliferation, cell cycle progression, apoptosis, DNA damages, ROS species production and the activation of repair pathways. We also preliminarily evaluated the cotreatment of PUN with TMZ, an already FDA-approved glioblastoma drug [27], to check if PDIA3 inhibition could be used as an adjuvant treatment for patients with glioblastoma.

## 2. Results

### 2.1. Inhibition and Knockdown of PDIA3 Decrease Cell Viability and Cell Proliferation in the T98G and U-87 MG Cell Lines

The effects of inhibiting PDIA3 were studied in terms of cell viability. The IC_50_ values for PUN in both T98G and U-87 MG cell lines were determined using an MTT-based assay, testing cells at various concentrations of PUN for up to 72 h. The reduction of MTT substrate was measured at 24, 48 and 72 h, and the PUN treatment resulted in a concentration- and time-dependent decrease in cell viability for both cell lines (Figure 1a). T98G cells were slightly more sensitive to PUN exposure than U-87MG cells. After 48 h of incubation with 30 μM PUN, less than 41% of the T98G cells retained viability, whereas this percentage was 54% for the U-87 MG cells. The IC_50_ values for PUN were extrapolated from experimentally obtained data using GraphPad Prism 8.0, as described in the Materials and Methods Section. For the T98G cell line, the IC_50_ value after 48 h of stimulation was 25 ± 6 μM, whereas for U-87 MG, it was 33 ± 7 μM.

Based on the results obtained from the viability assays, 30 μM of PUN was chosen for further cytotoxicity assays in both glioblastoma cell lines.

The growth of the T98G and U-87 MG cells after PDIA3 inhibition was monitored with a live cell microscope. Cultures were incubated with 30 μM of PUN for up to 48 h and images of cell growth (Figure 1b) were taken every 24 h. The results showed an antiproliferative effect of the PUN treatment on both glioblastoma cell lines, and these data were further confirmed by counting T98G and U-87 MG cells after the PUN treatment with a flow cytometer (Figure 1c).

We compared PDIA3 expression levels in both glioblastoma cell lines before and after 48 h PUN treatment. A Western blot analysis showed that the T98G cell line had approximately 35% higher PDIA3 levels, which could suggest a more extensive role of PDIA3 in T98G cells and a greater sensitivity to its inhibition, as demonstrated by the IC_50_ values. The PUN treatment increased PDIA3 levels in the U-87 MG cells by approximately 25%, whereas the increase in T98G cells was not significant (Figure 2a).

To further confirm the role of PDIA3 in glioblastoma, we examined the effect of PDIA3 knockdown on cell viability and proliferation in both glioblastoma cell lines. Firstly, the specificity of shRNA targeting PDIA3 was confirmed through an immunoblot analysis (Figure 2b), and we observed an approximately 50% decrease in PDIA3 expression levels. Consistent with the experimental data obtained for the PUN treatment, PDIA3 knockdown reduced cell viability and proliferation in both cell lines (Figure 2c,d) highlighting the role of PDIA3 in these processes. PDIA3-silenced T98G and U-87 MG cells were further incubated for 48 h with 30 μM of PUN, and no significant increase in cytotoxic effects was observed (Figure 2c,d), showing similar results to those obtained on mock-treated cells further incubated with PUN. These results suggested that the observed effects of PUN on cell viability and proliferation in the T98G and U-87MG cells were PDIA3-mediated. Additionally, transfection seemed not to affect PUN activity, because the PUN treatment on mock-treated and untransfected cells gave comparable results (Figure 1a,c and Figure 2c,d).

### 2.2. PDIA3 Inhibition Leads to PDIA3 Aggregation and Cell Cycle Arrest

It was previously reported by Medinas et al. [28] that a PDIA3 mutation with the loss of functions induced protein aggregation. Considering that PUN can inhibit PDIA3 enzymatic activity, we investigated if the PUN treatment could affect the cellular distribution of PDIA3 in both glioblastoma cell lines with immunofluorescence staining. A time course analysis revealed a change in the cellular distribution of PDIA3 in both cell lines after 16 h of stimulation with 30 μM of PUN, along with the appearance of PDIA3 aggregates (Figure 3). The effect became marked after 24 h and was more pronounced in the T98G cell line (Figure 3a, highlighted by arrows in the inset). In fact, PDIA3 aggregates in U-87 MG cells were smaller than those observable in T98G ones, and this could have been related to the difference between the two cell types and the higher level of PDIA3 expression observed in T98G cells. It seemed that in the U-87 MG cells, the aggregates at a late time point became smaller, and this could have been the result of some proteolytic activities.

Since the inhibition of PDIA3 was reported to affect cell cycle progression and apoptosis in MDA-MB-231 breast cancer and HCT116 colon cancer cells [29,30], and considering that the WHO grade IV gliomas have a high propensity for proliferation sustained by an efficient cell cycle [31], we investigated whether inhibiting PDIA3 could lead to a cell cycle progression arrest and eventually trigger apoptosis in glioblastoma cells.

T98G and U-87 MG cultures were treated with 30 μM of PUN for 16 and 48 h and analyzed with flow cytometry. The incubation with a final concentration of 5 μM camptothecin (CPT) for 16 h was used as a control for the cell cycle arrest. The results showed that PUN induced cell accumulation in the G_1_ phase, which was more evident in the U-87 MG cell line (Figure 4).

Consistent with the viability and cell proliferation results, the PUN-mediated PDIA3 inhibition induced cytostatic and cytotoxic effects through arresting cell cycle progression at the G_1_/S checkpoint. The cells underwent a series of surveillance mechanisms during the cell cycle to check their size, DNA replication, integrity and chromosome segregation [32]. PUN-treated T98G and U-87 MG cells showed an inefficient cell cycle due to a partial impairment in switching from the G_1_ to S phase of the cell cycle to duplicate their genomic DNA.

To test if the cell cycle arrest triggered by the PUN treatment could also induce apoptosis, an annexin V-base assay was conducted on the T98G and U-87 MG cells (Figure 5). After 16 h of treatment, a slight increase in the percentage of apoptotic cells was observed in both cell lines, which increased over time and was markedly higher after 48 h of treatment.

### 2.3. PDIA3 Inhibition Causes Activation of γ-H2AX and NHEJ Repair Pathway in a ROS-Independent Manner

To determine the mechanisms responsible for the accumulation of glioblastoma cells in the G_1_ phase following PDIA3 inhibition with the PUN treatment, we investigated the role of DNA damage and repair markers. We analyzed γ-H2AX and Ku70, markers for DNA damage and the nonhomologous end-joining (NHEJ) repair pathway, respectively [33]. We treated both glioblastoma cell lines with 30 μM of PUN and evaluated the expression of γ-H2AX and Ku70 through Western blotting. Both cell lines showed a time-dependent increase in γ-H2AX levels, which peaked between 4 and 6 h for the U-87 MG cells and at 8 h for the T98G cells (Figure 6a).

The PDIA3 inhibition also induced the upregulation of Ku70 levels in both cell lines, demonstrating the presence of double-strand DNA breaks. The heterodimer Ku70/80 is a key actor of the NHEJ repair pathway, as it rapidly recognizes broken DNA ends and protects them from nuclease activities [34]. However, the increase in the Ku70 expression levels peaked at 4–6 h of stimulation in the U-87 MG cells, while in the T98G cells, it was more significative and increased over time (Figure 6b). The higher and earlier activation of the NHEJ DNA repair pathway in the U-87 MG PUN-treated cells could explain the lower fold change observed in γ-H2AX expression as a result of better efficiency in repairing the DNA damage. This was in agreement with what was previously reported on the two cell lines used. In fact, while the U-87 MG cell line carries wild-type p53 [35], the T98G cells are characterized by a p53 mutation that blocks the transcriptional activity of the protein [36]. A comet assay confirmed the occurrence of DNA damage induced by the PDIA3 inhibition in both glioblastoma cell lines (Figure 6c).

We investigated whether the activation of γH2A.X upon the use of the PUN treatment was ROS-dependent by measuring the formation of ROS species using the CellROX^TM^ Green Reagent with flow cytometry. After 4 h from the administration of 30 μM of PUN in both glioblastoma cell lines, the PUN treatment showed no impact on the fluorescence of the probe for ROS species in both cell lines (Figure 7a). We also confirmed through the in vitro ABTS radical scavenging activity that PUN had antioxidant activity even at low micromolar concentrations, higher than ascorbic acid (Figure 7b). Thus, we concluded that the phosphorylation of H2A.X was not ROS-mediated and could be ascribed to PDIA3 inhibition through the use of the PUN treatment.

### 2.4. PDIA3 Inhibition as an Adjuvant for Temozolomide Chemotherapy Treatment in Glioblastoma

Considering that TMZ-resistant T98G cells were more sensitive to the PDIA3 inhibition, we determined the adjuvant action of PDIA3 inhibition on the treatment of glioblastoma with the chemotherapy agent TMZ. We first performed an MTT assay on both glioblastoma cell lines treated with increasing concentrations of TMZ (ranging from 30 μM up to 1 mM) for 24 and 48 h. Consistent with data in the literature, the T98G culture showed lower TMZ sensitivity compared to the U-87MG cells (Figure 8a).

We then extrapolated the lowest TMZ concentration that induced a decrease in cell viability (250 μM and 125 μM TMZ for T98G and U-87MG cell lines, respectively) for further experiments. Next, we examined the cell proliferation rate using a clonogenic assay, which has become the “gold standard” for assessing cellular sensitivity to chemotherapeutic agents. This test measures a single cell’s ability to undergo sufficient proliferation to form a colony. The T98G and U-87 MG cell lines were assayed in response to the PUN and TMZ treatments separately and to a PUN–TMZ cotreatment. The 10-day PUN–TMZ cotreatment showed lower colony formation in both GBM cultures compared to the single treatments, highlighting the synergic effect of PUN in the presence of TMZ on cell proliferation (Figure 8b).

## 3. Discussion

PDIA3, a multifunctional member of the PDI family with multiple cellular functions, has been linked to various types of cancers, neurological disorders and other diseases [37,38]. The role of PDIA3 in maintaining the correct redox state of disulfide bonds extends beyond ER resident proteins and has been observed in various cellular compartments, including the nucleus, cell membrane and mitochondria, highlighting its function as a molecular chaperone [39,40].

PDIA3 is overexpressed in several tumors [5,6,7], suggesting that tumor cells, and the associated oncogenic pathways, may require higher levels of PDIA3 to support the overexpression or activation of PDIA3-dependent processes compared to normal cells.

In recent years, there has been growing evidence of the potential therapeutic benefits of PDIA3 inhibition in various cancers, including clear cell renal cell carcinoma [41], drug-resistant ovarian cancer [42], breast cancer [43,44], hepatocellular carcinoma [45] and in multidrug-resistant gastric cancer [46]. In this study, we aimed to investigate the potential importance of PDIA3 in glioblastoma, one of the most aggressive forms of brain tumor, by examining the antiproliferative effects of PDIA3 inhibition in two glioblastoma cell line models (T98G and U-87 MG, TMZ-resistant and -sensitive, respectively).

In our two previous papers [25,26], we identified and characterized the inhibitory effect of PUN on PDIA3 enzymatic activity in vitro, proposing a binding site for PUN on a cleft of PDIA3 as the basis for drug optimization.

Here, we used PUN as a PDIA3 inhibitor in both glioblastoma cell lines and demonstrated that the inhibition of PDIA3 decreased the proliferation rate in these cultures. The same antiproliferative effects occurred using a PDIA3 shRNA vector that reduced the PDIA3 expression levels. The PDIA3 inhibition did not affect PDIA3 expression levels in the T98G cells and slightly increased PDIA3 levels in the U-87 MG cells, as confirmed through Western blot analyses, whereas it affected PDIA3 distribution in both glioblastoma cell lines. In fact, PDIA3 aggregates could be visualized with the use of immunofluorescence staining in both glioblastoma cell lines after PDIA3 inhibition. We correlated the cellular cytotoxicity induced by the PUN treatment, evidenced as a reduction in cell viability and proliferation, with the accumulation of such PDIA3 aggregates, probably represented by inactive proteins. Additionally, a cell cycle analysis showed an accumulation of cells in the G_1_ phase in both glioblastoma cell lines, indicating that deficiency in PDIA3 chaperone activity, caused by PDIA3 inhibition, directly affected DNA replication in the S phase of the cell cycle, leading to a cell cycle arrest in G_1_. PDIA3 has been shown to associate with APE/Ref-1, Ku80, Ku70 and the nuclear matrix protein 200/hPso4, all proteins involved in DNA repair mechanisms [47,48]. To investigate the causes of cell cycle arrest in the G_1_ phase, we examined DNA damage in both glioblastoma cultures induced using PUN. The ROS-independent phosphorylation of H2A.X was detected in both glioblastoma cell lines following the PUN treatment, which may represent an early cellular response against double-strand breaks of DNA [49]. γ-H2AX at the level of double-strand break sites alters the chromatin structure, opening a time window for the recruitment of repair factors [50]. Through a time course Western blotting analysis we also found the overexpression of the Ku70 repair factor, a marker for the activation of the NHEJ repair pathway, coinciding with the timeline of H2A.X phosphorylation. Furthermore, the presence of DNA damage was confirmed using comet assays in both glioblastoma cell lines. Additionally, the presence of DNA damage was evaluated in terms of proapoptotic stimuli by performing an annexin V assay. The results highlighted that PDIA3 inhibition increased the cell propensity to undergo apoptosis, a slightly more noticeable effect in the T98G cell line.

Based on the data obtained, PDIA3 might be a novel target for the treatment of glioblastoma, also in TMZ-resistant glioblastoma. Our preliminary clonogenic assays demonstrated a synergistic effect of the combined treatment, suggesting that it could be a strategy to enhance the effectiveness of the TMZ treatment. This finding indicated that targeting PDIA3 may provide a promising approach for improving the therapeutic outcomes in glioblastoma patients.

## 4. Materials and Methods

### 4.1. Cell Cultures

Human glioblastoma cell lines T98G and U-87 MG were obtained from the American Type Culture Collection (ATCC, Manassas, Virginia, USA). Cells were grown to 70% confluence at 37 °C in 5% CO_2_ in DMEM-HG medium (Sigma-Aldrich, Milan, Italy, cat. D5671) supplemented with 1% sodium pyruvate (Sigma-Aldrich, Milan, Italy, cat. S8636), 10% fetal bovine serum (FBS) (Sigma-Aldrich, Milan, Italy, cat. F7524), 2 mM glutamine (Sigma-Aldrich, Milan, Italy, cat. G7513) and 100 μg/mL streptomycin and 100 U/mL penicillin (Sigma-Aldrich, Milan, Italy, cat. P4333). Punicalagin (Sigma-Aldrich, Milan, Italy, cat. P0023, purity > 98%), dissolved in water, was tested on each cell line at different concentrations, from 1 μM up to 30 μM.

### 4.2. Viability Assay

T98G and U-87 MG cells were seeded into 96-well plates (8.000 cells/well) and PUN effect on cell viability at different concentrations was evaluated after 24, 48 and 72 h of incubation. The enzymatic reduction of MTT (3-(4,5-dimethylthiazol-2-yl)-2,5-diphenyl-2H-tetrazolium bromide) (Sigma-Aldrich, Milan, Italy, cat. M2128) to MTT formazan was used to assess cell viability. Cell cultures were further incubated for 2 h in culture medium containing 0.5 mg/mL MTT. After 2 h of incubation, the solution was removed and the blue MTT formazan product was dissolved in DMSO (Sigma-Aldrich, cat. D8418). After 30 min at room temperature, the absorbance of the formazan solution was spectrophotometrically read at 570 nm using an Appliskan^®^ plate reader (Thermo Fisher Scientific, Monza, Italy).

### 4.3. PDIA3 Knockdown Procedure

PDIA3-silenced cells were obtained using PDIA3 shRNA vector (Mission shRNA library, Sigma Sigma-Aldrich, Milan, Italy) and Lipofectamine 2000 (Thermo Fisher Scientific) according to manufacturer’s instructions. Briefly, both glioblastoma cell lines were seeded into 6-well plates (240,000 cells/Well); 24 h after seeding, 4 μg PDIA3 shRNA vector as well as 10 μL Lipofectamine 2000 were separately diluted in 250 μL FBS- and antibiotic-free DMEM-HG medium and incubated for 5 min. After incubation, the diluted PDIA3 shRNA vector with the diluted Lipofectamine 2000 were combined and incubated for extra 20 min. In total, 500 μL PDIA3 shRNA vector–Lipofectamine 2000 complexes were added to 1 mL antibiotic-free DMEM HG medium to each well. Mock-treated cells were incubated with an empty plasmid. Cells were transfected for 48 h and further analyzed for cell viability and immunoblotting analyses.

### 4.4. Cell Imaging

Glioblastoma cell lines were seeded into 24-well plates (50,000 cells/Well). Incucyte live cell imaging system (Sartorius, Milan, Italy) was used for showing cell proliferation and morphology. The system took a photo of cell plates (using the same spatial coordinates) every 24 h in a bright field channel.

### 4.5. Cell Counting

Cells were detached using trypsin/EDTA (Sigma-Aldrich, Milan, Italy, cat. T4049) harvested through centrifugation and resuspended in DMEM-HG medium. Cells were gently inverted 10 times before analysis and immediately analyzed with Accuri™ C6 flow cytometer (BD Biosciences, Milan, Italy). Samples were analyzed at 35 μL/s rate (<500 events per second) using an FSC-H threshold of 25,000 to exclude debris and electronic noise. A 100 μL aliquot was collected for each sample and appropriate gates were drawn around cell population. The number of cells for each sample was expressed as a percentage of the control.

### 4.6. Protein Extraction and Immunoblotting Analysis

Cells cultured were scraped, harvested through centrifugation, washed in phosphate-buffered saline, PBS (Sigma–Aldrich, Milan, Italy, cat. D8662) and lysed in buffer containing 2% SDS (BioRad, Segrate (MI), Italy, cat. 161030), 20 mM tris-hydrochloride pH 7.4 (Sigma–Aldrich, Milan, Italy, cat. T3253), 2 M urea (Sigma–Aldrich, Milan, Italy, cat. U5378), 10% glycerol (Merck, Milan, Italy, cat. GE17-1325-01) supplemented with 2 mM sodium orthovanadate (Sigma–Aldrich, Milan, Italy, cat. S6508), 10 mM DTT (Sigma–Aldrich, Milan, Italy, cat. D9779) and a protease inhibitor cocktail diluted to 1:100 (Immunological Sciences, Roma, Italy, cat. IK-96010). Proteins resolved using SDS-PAGE 10% TGX FastCast Acrylamide gel (BioRad, Segrate, Italy, cat. 161-0183) were transferred on polyvinylidene fluoride (PVDF) membranes (BioRad, cat. 1620174) using Trans-Blot^®^ Turbo^TM^ Transfer System (BioRad, cat. 170-4247). The membranes were blocked with 0.2% w/v I-Block^TM^ (Invitrogen, Monza, Italy, cat. T2015) in tris-buffered saline (TBS, 20 mM Tris, 150 mM NaCl, pH 7.6, all chemicals were from Sigma-Aldrich) and subjected to immunoblotting using specific primary antibodies for 1 h. Anti-Ku70 (Santa Cruz Biotechnology, Segrate (MI), Italy, cat. sc-9033), anti-β-actin (Sigma-Aldrich, cat. A1978 clone AC-15) and Alexa Fluor^®^ 647 antigamma H2A.X (phospho S139) (Abcam, Cambridge, UK, cat. 195189) were used for immunoblotting detections. The membranes were then washed three times in I-Block^TM^ 0.2% *w*/*v* in TBS and incubated for 1 h with the appropriate alkaline phosphatase (Sigma-Aldrich, cat. A3687 and A3688, dilution 1:5000) or peroxidase (Jackson Immuno Research, Cambridgeshire, United Kingdom, cat. 115-035-174, dilution 1:5000) to conjugatd secondary antibodies. The alkaline phosphatase signal was detected with BCIP/NBT reagents (Carl Roth, Milan, Italy, CAS no. 298-83-9 and 6578-06-9). The peroxidase signal was detected with ECL Fast Femto reagent (Immunological Sciences, Rome, Italy, cat. 34094). β-actin was used as a normalization protein. At least three experimental immunoblotting detections were performed for each biological sample.

### 4.7. Cell Cycle Analysis

A propidium-iodide-based staining of DNA content was used to measure the percentage of cells in each cell cycle phase (G_0_/G_1_, S and G_2_/M). Briefly, cell cultures were treated with 30 μM PUN for 16 and 48 h. Then, the cells were detached with trypsin/EDTA, harvested through centrifugation, washed in HBSS (Sigma Aldrich, Milan, Italy, cat. 55021C) and fixed with 70% cold ethanol. Ethanol was added dropwise to the pellet while mixing through inversion. After 30 min incubation at 4 °C, the cells were washed twice in HBSS buffer, and 0.2 mg/mL RNAse (Sigma Aldrich, cat. R6513) was added for 5 min at 37 °C. Then, 60 μg/mL propidium iodide (Sigma Aldrich, cat. P4170) was added and cells were incubated for 45 min at 37 °C in the dark. Before flow cytometry analysis, samples were centrifuged and resuspended in HBSS. Samples were analyzed using an Accuri™ C6 flow cytometer (BD Biosciences, Milan, Italy).

### 4.8. Clonogenic Potential Assay

The ability of the glioblastoma cells to generate in vitro colonies was determined using clonogenic assay. Briefly, T98G and U-87 MG cells were plated into 6-well plates (1000 cells/well) and grown until colony formation. A total of 24 h after seeding, 5 μM PUN and TMZ (125 μM for U-87 MG and 250 μM for T98G cells) was added to both cell lines as single treatments and in cotreatment. After 10 days, supernatants were removed and the colonies were gently washed with PBS (Sigma–Aldrich, cat. D8662), fixed with cold methanol for 20 min and stained with 0.1% crystal violet in methanol/PBS (ratio 1:4) at room temperature for 1 h. Then, colonies were washed in PBS and air-dried. Images were captured and the total area of colonies/well was measured using Image J software (free software, website: https://imagej.nih.gov/ij/ accessed on 18 July 2023).

### 4.9. Immunofluorescence

Immunofluorescence analysis was performed according to Cocchiola et al. [51]. Cultured cells were seeded on coverslips and stimulated for 16 and 48 h with 30 µM of PUN. Cells grown on coverslips were washed with PBS (Sigma–Aldrich, cat. D8662), fixed with 3.7% formaldehyde for 20 min and then rinsed with PBS. Cells were permeabilized with cold methanol (− 20 °C) for 5 min. After washing three times with PBS, the cells were blocked for 1 h with 3% *w*/*v* BSA (Immunological Sciences, Roma, Italy, cat. ISP6154-100) in PBS. Fixed cells were processed with immunofluorescence staining using specific primary antibodies against PDIA3 (Millipore, Milan, Italy, cat. ABE1032) properly diluted in PBS containing 2% *w*/*v* BSA (Sigma-Aldrich, cat. A5611) for 1 h. Cells were washed three times with 2% *w*/*v* BSA in PBS and then incubated for 1 h in darkness with an AlexaFluor 488-conjugated secondary antibody (Jackson Immunoresearch, Cambridge, UK, cat. 211-545-109, dilution 1:1000). Nuclei were counterstained with 100 ng/mL Hoechst (Sigma–Aldrich, Milan, Italy, cat. 94403) for 20 min. After washing with PBS + 0.05% Triton, coverslips were mounted on glass microscope slides with Duolink^TM^ mounting medium and examined using a fluorescence microscope (Leica AF6000 Modular System, Leica, Milan, Italy) with 63× oil immersion objective. Samples were captured under the same acquisition parameters.

### 4.10. Comet Assay

T98G and U-87 MG cells were seeded on a 6-well plate (240,000 cells/well) and treated with 30 μM of PUN for 8 and 4 h, respectively. Cells stimulated with 100 μM tert-butyl hydroperoxide (tBHP, from Sigma-Aldrich) for 1 h were used as a positive control for DNA fragmentation. Cells were detached with trypsin, neutralized, centrifugated and resuspended at a ratio of 1:10 (*v*/*v*) in 1% low-melting-point agarose (Millipore, cat. 2070 OP). In total, 20 μL of the cells/agarose mixture was pipetted onto a glass slide (preliminarily coated with 1% agarose (Sigma–Aldrich, Milan, Italy, cat. A0576)) and incubated overnight in a lysis solution (2.5 NaCl, 100 mM EDTA, 10 mM tris-base, 200 mM NaOH, 1% sodium lauryl sarcosinate and 1% Triton X-100, all chemicals were from Sigma-Aldrich) at 4 °C. Then, slides were immersed for 1 h at 4 °C in alkaline electrophoresis solution pH > 13 (200 mM NaOH, 1 mM EDTA, all chemicals were from Sigma-Aldrich) to allow DNA unwinding. Electrophoresis was carried out at 40 mA for 1 h at 4 °C. Then, slides were rinsed twice with dH_2_O and immersed in 70% ethanol for 30 min at room temperature. After drying the slides for 15 min at 37 °C in the dark, slides were stained with propidium iodide (Sigma–Aldrich, Milan, Italy, cat. P4170) at a final concentration of 5 μM/mL (15 min at room temperature); then, slides were rinsed with dH_2_O and dried for the acquisition. Images were captured using a fluorescence microscope (Leica AF6000 Modular System) with 20× objective.

### 4.11. CellROX^TM^ Green Flow Cytometry Assay

ROS species were detected in T98G and U-87 MG cells upon 30 μM PUN treatment for 4 h using CellROX^TM^ Green reagent probe (Thermo Fisher Scientific, cat. C10492) according to the manufacturer’s instructions. Incubation with 100 μM tert-butyl hydroperoxide (TBHP, from Sigma-Aldrich) for 1 h was used as a positive control for the ROS species production. Samples were processed using an Accuri™ C6 flow cytometer (BD Biosciences) and results were analyzed using FCS Express 7 software (De Novo Software, Pasadena, CA, USA).

### 4.12. ABTS Radical Scavenging Assay of Punicalagin

To obtain an 2,2′-azino-bis (3-ethilbenzothiazoline-6-sulfonic acid (ABTS) radical cation (ABTS•+) solution, 7 mM ABTS and 2.45 mM potassium persulfate (both from Sigma-Aldrich, Milan, Italy) were incubated in the dark at room temperature for 12–16 h before use. Since PUN is water-soluble, ABTS•+ was diluted with PBS, pH 7.4, to an absorbance of 0.70 (±0.02) at 690 nm and equilibrated at room temperature. A total of 10 μL of each PUN sample at various concentrations (ranging from 0.15 up to 300 μM) was added to 190 μL of prediluted ABTS•+ solution. The reaction mixtures were incubated at room temperature for 10 min and immediately read at 690 nm with an Appliskan^®^ plate reader. ABTS radical-scavenging activity of PUN was expressed as inhibition percentage of ABTS•+ radicals. The IC50 value was defined as the PUN concentration at which 50% of ABTS•+ radicals was scavenged. L-ascorbic acid at various concentrations was used as positive control in the assay.

### 4.13. Statistics

All statistical analyses were performed with GraphPad Prism 8.0 software (GraphPad Software, San Diego, CA, USA). Comparisons between two groups for statistical significance were assessed using a two-tailed Student’s *t*-test. Statistical analyses on multiple groups were performed using ANOVA, followed by Tukey’s post hoc test. A *p*-value < 0.05 was considered statistically significant and all data presented were the result of at least three independent experiments.

## 5. Conclusions

The overexpression of PDIA3 is common in many types of cancers, and its inhibition offers a valid alternative for new treatments. In this paper, we demonstrated the efficacy of the PDIA3 inhibitor in terms of its cytotoxicity effects in two cellular models of glioblastoma, T98G and U-87 MG cell lines; based on the data obtained, the antiproliferative effects induced by PDIA3 inhibition were related to an increase in DNA damage. These preliminary findings offer promising prospects for the development of new therapeutic approaches for the treatment of glioblastoma. Further research in vivo and clinical studies are warranted to validate the efficacy and safety of PDIA3 inhibition in combination with TMZ, with the ultimate goal of improving patient outcomes in the WHO grade IV gliomas.

## Figures and Tables

**Figure 1 ijms-24-13279-f001:**
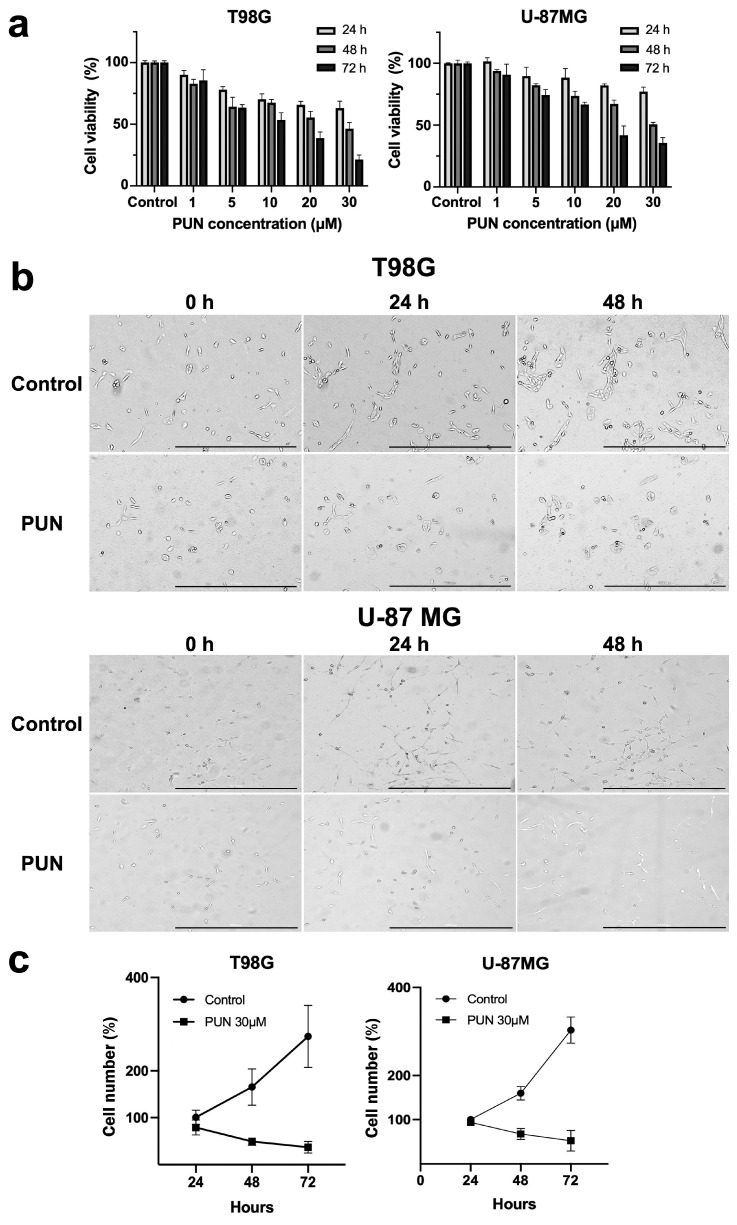
Inhibition of PDIA3 decreased cell viability and proliferation in T98G and U-87 MG cells: (**a**) MTT assays for T98G and U-87 MG cell lines. Cells were treated with PUN (from 1 up to 30 μM) for 24, 48 and 72 h. Control untreated cells were analyzed at the same time points. Experiments were repeated three times and the obtained results were reported as the mean and SEM. (**b**) Representative bright field images of T98G and U-87 MG cells incubated with 30 μM PUN for 24 and 48 h (scale bar 1000 µm). Control untreated cells were analyzed at the same time points. Images were taken every 24 h. (n = 2) (**c**) Growth curve of 30 μM PUN-treated glioblastoma cells incubated for 24, 48 and 72 h. Control untreated cells were analyzed at the same time points. Values were expressed as the mean and SEM (n = 3).

**Figure 2 ijms-24-13279-f002:**
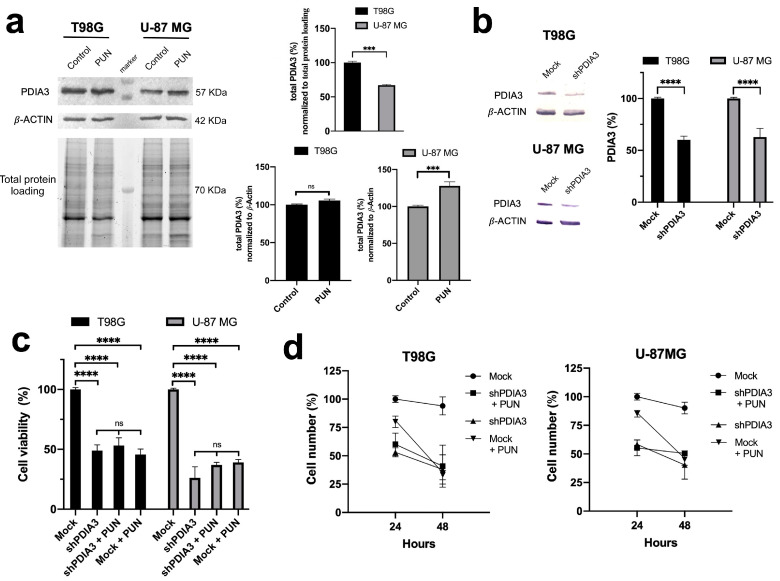
Effects of PUN stimulation on PDIA3 expression and PDIA3 knockdown on cell viability and proliferation: (**a**) Western blot analysis (left panel) and relative densitometric analysis (right panel) of T98G and U-87 MG cells treated with 30 μM PUN for 48 h. Total protein loading was used to normalize PDIA3 levels between T98G and U-87 MG cells, while *β*-actin levels were used to normalize treated cells with respect to control (untreated cells incubated for 48 h). Values were expressed as mean and SEM (n = 3). Statistical analysis was performed using a two-tailed Student’s *t*-test comparing all data with the control (ns, not statistically significant, *** *p*  <  0.001). (**b**) Western blot analysis (left panel) and relative densitometric analysis (right panel) of T98G and U-87 MG cells PDIA3-silenced for 48 h. Mock (empty plasmid) was used as a control (see Materials and Methods Section for experimental protocols). Values, shown as mean and SEM (n = 4), were normalized against *β*-Actin and expressed as percentage of mock. Statistical analysis was performed using a two-tailed Student’s *t*-test comparing all data with mock (**** *p*  <  0.0001). (**c**) MTT assays were performed on PDIA3-silenced T98G and U-87 MG cells untreated or treated with 30 μM of PUN. Mock and mock plus 30 μM of PUN were used as a control. Values were expressed as mean and SEM (n = 4). Statistical analysis was performed using a two-tailed Student’s *t*-test comparing all data with the control (ns, not statistically significant, **** *p* <  0.0001). (**d**) Growth curves for PDIA3-silenced T98G and U-87 MG cells untreated or treated with 30 μM of PUN for 24 and 48 h. Mock and mock plus 30 μM of PUN were used as a control. Values were expressed as mean and SEM (n = 4).

**Figure 3 ijms-24-13279-f003:**
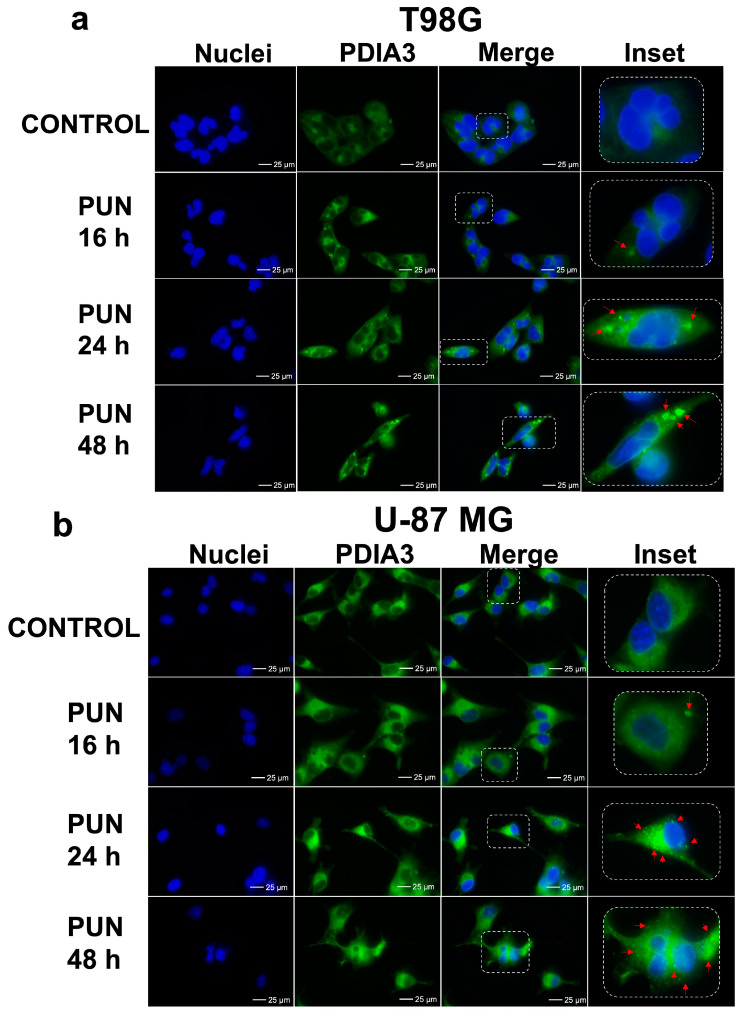
Cellular aggregates of PDIA3 upon PUN stimulation visualized through immunofluorescence staining of PDIA3: T98G (**a**) and U-87 MG (**b**) cells were subjected to 30 μM PUN treatment for 16 h, 24 h and 48 h. PDIA3 aggregates were visible after 16 h stimulation, with more evidence after 24 h and 48 h. Control corresponded to untreated cells. Images reported were representative of three independent experiments and were captured under the same acquisition parameters. Insets show enlarged portions of the acquired images, and the red arrows point to PDIA3 aggregates. Images were captured with a confocal fluorescence microscope (Leica AF 6000) using 60× oil immersion objective (scale bar 25 μm).

**Figure 4 ijms-24-13279-f004:**
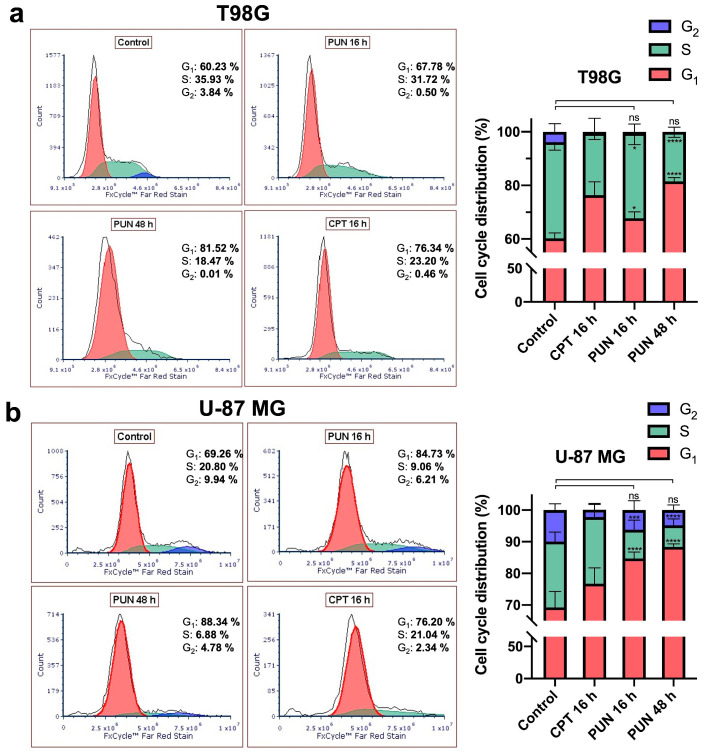
Cell cycle analysis performed on T98G and U-87 MG cultures upon pharmacological inhibition of PDIA3: (**a**) cell cycle distribution profile in T98G cells treated with 30 μM of PUN for 16 and 48 h and stained with FxCycle^TM^ Far Red Stain compared to untreated cells (control). A final concentration of 5 μM camptothecin (CPT) for 16 h was used as control for cell cycle arrest. PUN-treated cells showed a tendency to reduce S phase, abolish G_2_ phase and enrich in G_1_ phase (left panel). (**b**) U-87 MG cells, treated as T98G cells, showed a tendency to reduce S and G_2_ phases and a time-dependent enrichment in G_1_ phase (left panel). The histograms shown in the right panels summarized the cell percentage in the different phases. Data were analyzed with FCS Express 7, shown as mean and SD, and were representative of three independent measurements. Statistical analyses were performed with GraphPad software using ANOVA, followed by Tukey’s post hoc test for comparisons of multiple groups (ns, not statistically significant, * *p* < 0.05, *** *p* < 0.001, **** *p* < 0.0001).

**Figure 5 ijms-24-13279-f005:**
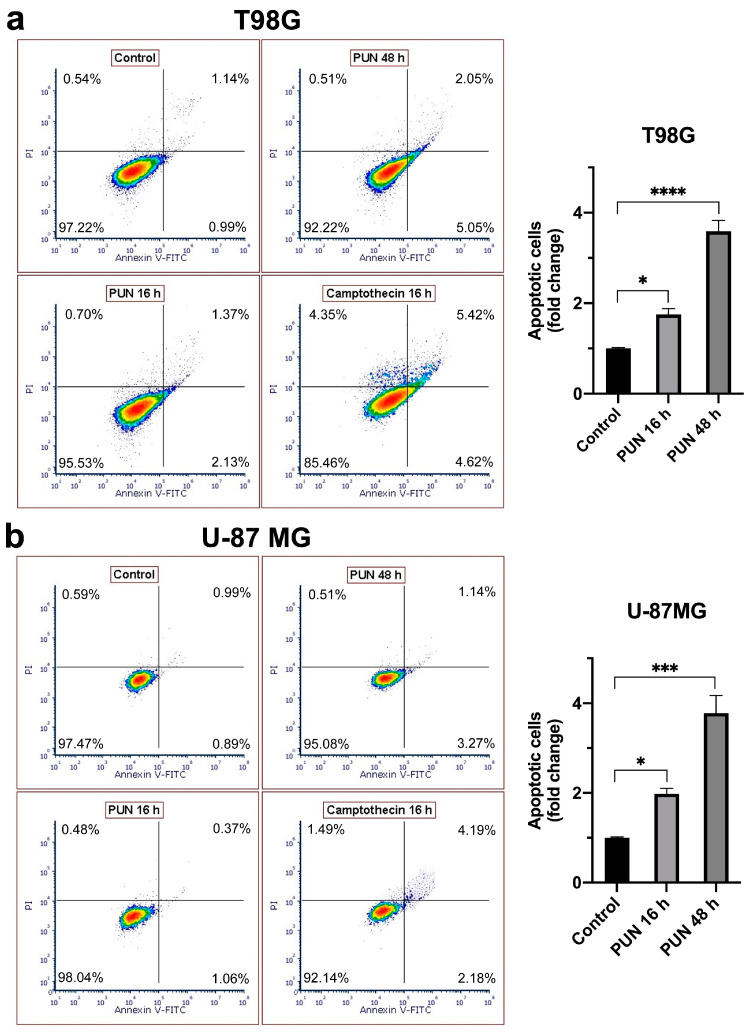
Annexin V apoptosis assays on T98G and U-87 MG cells upon PUN treatment: (**a**) T98G and (**b**) U-87MG cells were treated with 30 μM of PUN for 16 and 48 h and compared to untreated cells (control). Incubation with 5 μM camptothecin (CPT) for 16 h was used to promote apoptosis. Cells were stained with annexin V-FITC and propidium iodide (PI). Samples were analyzed through flow cytometry (at least 10,000 cells were selected using the same gating strategy). Apoptotic cells were considered as the sum of annexin V-FITC^+^/PI^−^ (early apoptosis cells) and annexin V-FITC^+^/PI^+^ (late apoptosis cells). Data were summarized in the two histograms on the right panels, normalized to the untreated control and expressed as mean values and SD (n = 3). Statistical analyses were performed with GraphPad software using ANOVA, followed by Tukey’s post hoc test for comparisons of multiple groups (* *p* < 0.05, *** *p* < 0.001, **** *p* < 0.0001).

**Figure 6 ijms-24-13279-f006:**
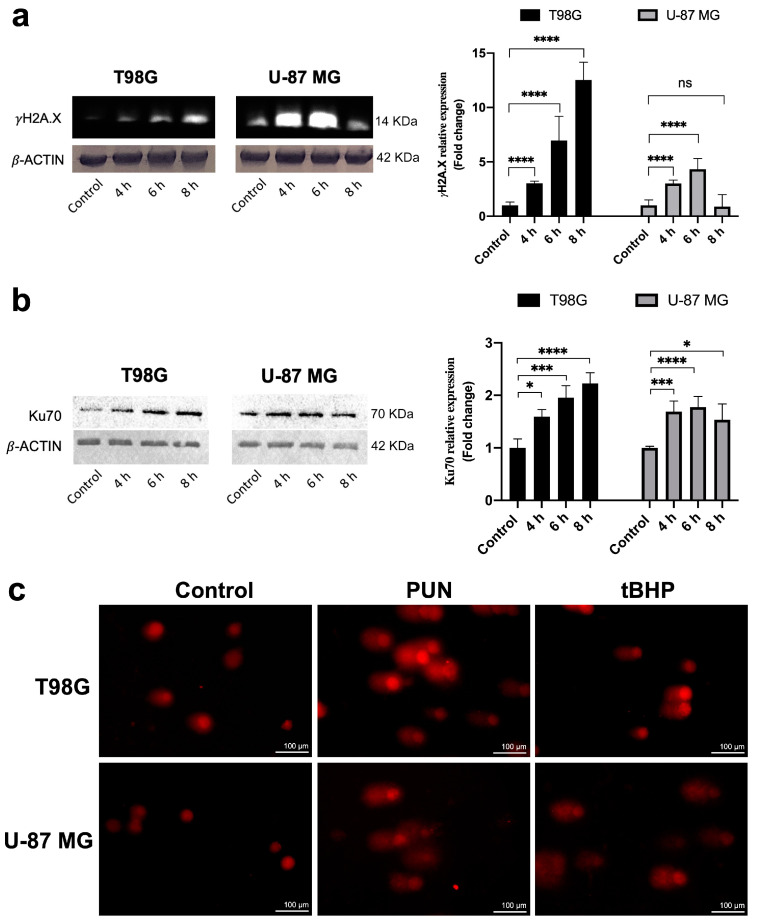
PDIA3 inhibition induced DNA damage and triggered the NHEJ repair pathway in T98G and U-87 MG cells: (**a**) representative Western blotting analysis (left panel) of γH2A.X on T98G and U-87 MG cells treated with 30 μM of PUN for 4, 6 and 8 h compared with untreated (control) cells. (**b**) Western blotting analysis (left panel) of Ku70 performed on T98G and U-87 MG cells treated with 30 μM of PUN for 4, 6 and 8 h compared with untreated (control) cells. In both cases, *β*-actin levels were used to normalize data. The right panels show the densitometric analysis of γH2A.X from the immunobloKts shown on the left side. Data were reported as the mean of the fold change and SD (n = 3). Statistical analyses were performed using a two-tailed Student’s *t*-test comparing each time point with the control (ns, not statistically significant, * *p* < 0.05, *** *p* < 0.001, **** *p* < 0.0001). (**c**) Comet assay was performed to evaluate DNA damage. T98G and U-87 MG were treated with 30 μM of PUN for 8 and 4 h, respectively, compared with untreated (control) cells. As shown in the figure, PUN treatment caused DNA fragmentation in both glioblastoma cell lines. A total of 100 µM tBHP was used for 1 h as a positive control for DNA fragmentation in both glioblastoma cell lines. Experiments were repeated 3 times with similar results.

**Figure 7 ijms-24-13279-f007:**
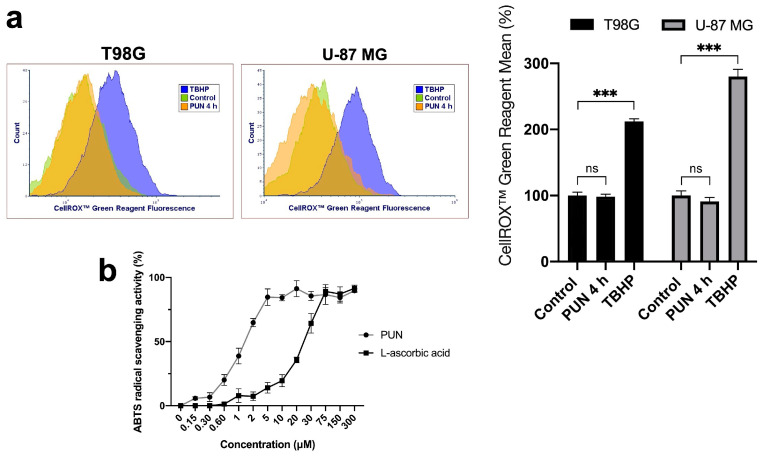
ROS species analysis through CellROX^TM^ Green Reagent assay for PUN-treated T98G and U-87 MG cells and ABTS radical scavenging assay for PUN compound: (**a**) glioblastoma cell lines were incubated in the absence (control) and the presence of 30 μM of PUN for 4 h, and CellROX^TM^ fluorescence probe was added for extra 30 min of incubation. Incubation with 100 μM TBHP for 30 min was used as positive control for the generation of ROS species. No shift in terms of CellROX^TM^ Green Reagent fluorescence was observed upon 4 h of PUN treatment in both glioblastoma cell lines (left panel), and the histogram (right panel) summarized the change in ROS species upon different treatments. Statistical analyses were performed with GraphPad software using ANOVA, followed by Tukey’s post hoc test for comparisons of multiple groups (ns, not statistically significant, *** *p* < 0.001). (**b**) ABTS assay was performed to prove the antioxidant nature of PUN; L-ascorbic acid was used as an antioxidant standard compound. PUN exhibited in vitro net antioxidant properties.

**Figure 8 ijms-24-13279-f008:**
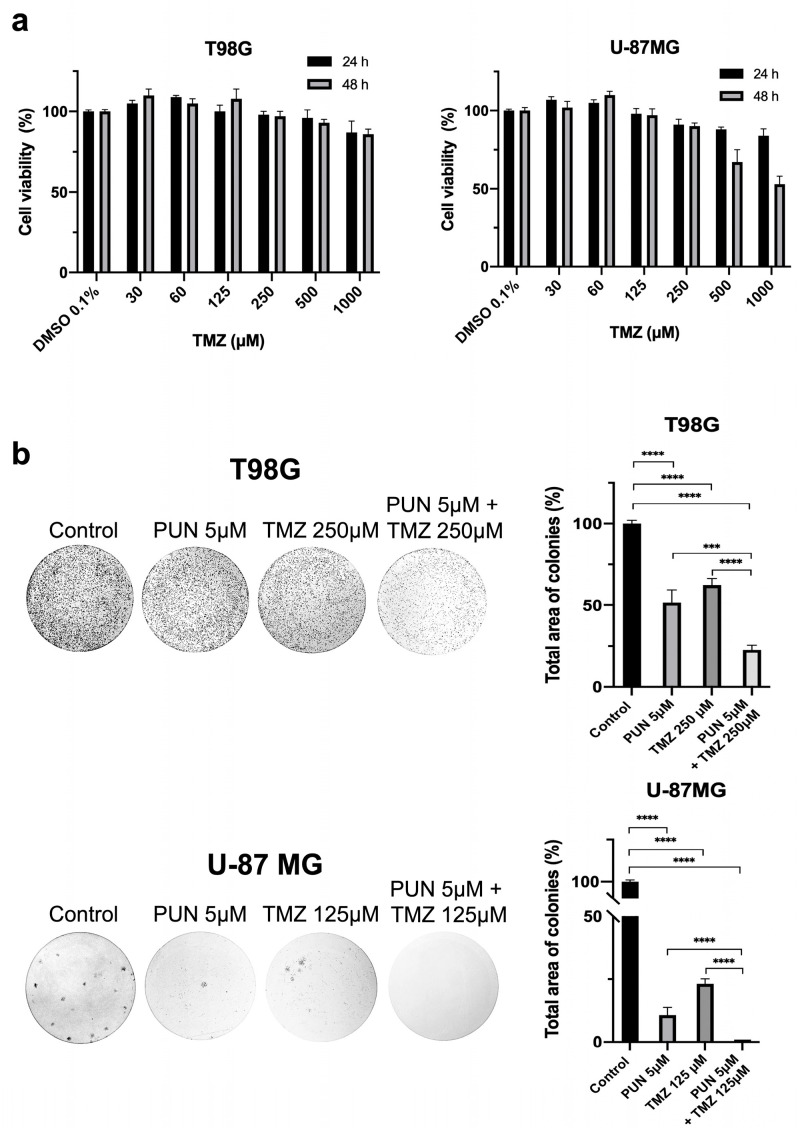
Synergistic effect of the PUN–TMZ cotreatment in decreasing the clonogenic potential in T98G and U-87 MG cells: (**a**) MTT assays on T98G and U-87 MG cultures treated with increasing TMZ concentrations, ranging from 30 to 1000 μM, for 24 and 48 h. DMSO 0.1% was used as the control. Values were expressed as mean and SEM (n = 3). (**b**) Representative clonogenic assays conducted on T98G and U-87 MG cell lines. Both cells were treated with PUN (5 μM) or TMZ (250 μM and 125 μM for T98G and U-87 MG cells, respectively) and cotreated with PUN–TMZ (using the same concentrations of single treatments). Untreated cells were used as control. After the 10-day treatment, the colonies formed were evidenced using crystal violet dye (left panel), counted and total area of colonies (expressed as mean percentage with respect to control and SD) is shown in the histogram (right panel). The images reported are representative of three independent experiments. Statistical analyses were performed using ANOVA, followed by Tukey’s post hoc test for comparisons of multiple groups (*** *p* < 0.001, **** *p* < 0.0001).

## Data Availability

Data are contained within the article.

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
