# Peer review of "Protein Disulfide Isomerase A3 (PDIA3): A Pharmacological Target in Glioblastoma?"

_ijms, 2023, doi:10.3390/ijms241713279_

Round 1

Reviewer 1 Report

Paglia et al. described the influence of PDIA3 for glioblastoma cells in vitro. It seems an interesting potential target for novel therapeutic strategies because it reduced glioma cell proliferation and increased apoptosis in a ROS-independent manner.

Line numbers are missing. Difficult to refer comments to the the respective parts.

Major concerns

1.       Results 2.1. Second paragraph: “We observed a net loss of cytotoxic effect of 146 PUN treatment in PDIA3-silenced cells (Figures 2C and 2D).” There is no difference between sh and sh+PUN. Please explain more in detail what you mean.

2.       Fig. 2c: control Mock+PUN is missing. Please include.

3.       If you use different time points (Fig. 3, Fig. 4, Fig. 5, Fig. 6), the used time point of control is not clear. Please indicate.

4.       Fig. 3: Difficult to see the aggregates. Higher resolution is required. It seems that aggregates at late time point (U87) are smaller than at earlier time points? Explain or discuss.

5.       Fig. 5: The dot plots are not optimal. You have to compensate in an accurate way that you see defined populations. And you have to acquire more cells per sample (especially U87, data are not reliable if you have only 10 cells as target cells, or you have to use a control to exclude that this is only background). Use for depiction of percentages dots instead of commas. Did you perform a positive control for apoptosis induction?

6.       Fig. 8b: This doesn´t look like a colony formation assay for T98G. It looks as a cell layer. Maybe, you can take better pictures. Axis – “total area” but in the figure legend you describe colony numbers. Please adapt.

Minor concerns

1.       Last sentences of abstract AND Introduction: avoid overstatement! You only analyzed the effect of PUN in vitro. More experiments also in vivo are required to validate this potential. Revise.

2.       Introduction: Supplementary S1. This should be part of the results. Furthermore, you described Suppl. S1 as a survival curve but only expression level is shown.

3.       If you only take pictures every 24h, this is not really live cell imaging. Please revise.

4.       Results 2.1. Second paragraph: “the key role of PDIA3” – Overstatement. Revise.

5.       Results 2.1. Second paragraph: “are mainly PDIA3-mediated.” – Overstatement. Revise.

6.       Fig. 1b: More contrast would be helpful to see the cells. Please adapt.

7.       Fig. 1a/c: Please include statistics. Check axis.

8.       Fig. 2a: Graphs are too small. Labeling of axes are not readable.

9.       For WBs: include sizes for each protein in the figure or figure legend.

10.   Fig. 2: You directly start with (a). Heading is missing.

11.   Fig. 4: Histograms require a higher resolution. In the main text Camptothecin is not mentioned. Why do you use that reagent? Figure legend – define CPT.

12.   Fig. 5 Figure legend: please revise – histograms = graphs, they are not on the right.

13.   Fig. 6A: Why did you use white bands and black bands? Please unify.

14.   Discussion (before last paragraph): The shown induction of apoptosis is not “pronounced”. Please revise.

Only minor editing of language is required.

Author Response

Response to Reviewer 1

Paglia et al. described the influence of PDIA3 for glioblastoma cells in vitro. It seems an interesting potential target for novel therapeutic strategies because it reduced glioma cell proliferation and increased apoptosis in a ROS-independent manner.

Line numbers are missing. Difficult to refer comments to the the respective parts.

We apologize for the missing line numbers, but they were present in the manuscript we submitted. We are grateful to the reviewer for the valuable comments which helped to improve the quality of the manuscript.

Point by point response

Major concerns

  1. Results 2.1. Second paragraph: “We observed a net loss of cytotoxic effect of 146 PUN treatment in PDIA3-silenced cells (Figures 2C and 2D).” There is no difference between sh and sh+PUN. Please explain more in detail what you mean.

We observed that the further addition of PUN to PDIA3-silenced cells does not increase the cytotoxic effect. Based on this result, we can suppose that the observed effects of PUN on cell viability and proliferation in T98G and U-87MG cells are PDIA3-mediated. The manuscript has been modified as follows:

“PDIA3-silenced T98G and U-87 MG cells were further incubated for 48 hours with 30 μM PUN and no significant increase in cytotoxic effects was observed (Figures 2C and 2D), showing similar results to those obtained on mock-treated cells further incubated with PUN. These results suggest that the observed effects of PUN on cell viability and proliferation in T98G and U-87MG cells are PDIA3-mediated. Additionally, transfection seems not to affect PUN activity because PUN treatment on mock-treated and untransfected cells gave comparable results (Figures 1A,C and 2C,D).”

  1. Fig. 2c: control Mock+PUN is missing. Please include.

Control mock+puni has been included. Additionally, transfection seems not to affect PUN activity because PUN treatment on mock-treated and untransfected cells gave comparable results.

  1. If you use different time points (Fig. 3, Fig. 4, Fig. 5, Fig. 6), the used time point of control is not clear. Please indicate.

Figure legends have been modified to better explain each control used.

  1. Fig 3: Difficult to see the aggregates. Higher resolution is required. It seems that aggregates at late time point (U87) are smaller than at earlier time points? Explain or discuss.

We did not acquired images with a confocal microscope and resolution is limited by objective and camera used. We provided for each time point an inset with an enlarged image, but any further zoom may result in pixelization. However, we think that images provided support the presence of a dramatic change in PDIA3 cellular distribution that we interpretate as protein aggregate formation, based also on literature data. We agree with reviewer that aggregates in U87 cells are smaller than those observable in T98 ones and this could be related to the difference between the two cell types and the higher level of PDIA3 expression observed in T98 cells. We did not extend the observation beyond 48h, but it is possible that some of the aggregates could be cleared by proteolytic activities.

Manuscript has been modified by adding the following sentences:

“In fact, PDIA3 aggregates in U87 cells are smaller than those observable in T98 ones and this could be related to the difference between the two cell types and the higher level of PDIA3 expression observed in T98 cells. It seems that in U87 cells the aggregates at late time point become smaller and this could be the result of some proteolytic activities.”

  1. Fig. 5: The dot plots are not optimal. You have to compensate in an accurate way that you see defined populations. And you have to acquire more cells per sample (especially U87, data are not reliable if you have only 10 cells as target cells, or you have to use a control to exclude that this is only background). Use for depiction of percentages dots instead of commas. Did you perform a positive control for apoptosis induction?

Cytometric analyses reported in figures 4 and 5 have been performed on an equivalent number of cells (at least 10000 events) using the same gating strategy and avoiding collecting doublets. We also treated both cell lines with Camptothecin to induce apoptosis and these data have been provided in the revised manuscript.

  1. Fig. 8b: This doesn´t look like a colony formation assay for T98G. It looks as a cell layer. Maybe, you can take better pictures. Axis – “total area” but in the figure legend you describe colony numbers. Please adapt.

We agree with reviewer, but at present we do not have a better picture. However we trust that data provided support the conclusion drawn and a new cell colony experiment would require an additional time. Analysis was performed with ImageJ software, and we considered the total area of colonies (measured by the program) to avoid underestimation due to colonies overlap. Figure legend has been modified.

Minor concerns

  1. Last sentences of abstract AND Introduction: avoid overstatement! You only analyzed the effect of PUN in vitro. More experiments also in vivo are required to validate this potential. Revise.

Manuscript has been revised as follows:

Abstract

“Although further in vivo studies are needed, results obtained suggest PDIA3 as a novel therapeutic target that can be also included in already approved therapies.”

Introduction

“We also preliminarily evaluated the co-treatment of PUN with TMZ, an already FDA-approved glioblastoma drug [27], to check if PDIA3 inhibition could be used as an adjuvant treatment for patients with glioblastoma.”

  1. Introduction: Supplementary S1. This should be part of the results. Furthermore, you described Suppl. S1 as a survival curve but only expression level is shown.

We prefer to leave this part as supplementary because is not strictly experimental, but we can add it to the manuscript if required. We modified the text by adding this sentence: “We also analyzed mRNA expression of PDIA3 by using a gene expression profile downloaded from the GEO database and we found that PDIA3 is upregulated in glioblastoma than in normal brain tissue (see supplementary data).”.

  1. If you only take pictures every 24h, this is not really live cell imaging. Please revise.

We agree with reviewer, and we modified the Materials and Methods section as follows:

4.4 Cell imaging:

  1. Results 2.1. Second paragraph: “the key role of PDIA3” – Overstatement. Revise.

Manuscript has been revised as follows: “Consistent with the experimental data obtained for PUN treatment, PDIA3 knockdown reduced cell viability and proliferation in both cell lines (Figures 2C and 2D) highlighting the role of PDIA3 in these processes.

  1. Results 2.1. Second paragraph: “are mainly PDIA3-mediated.” – Overstatement. Revise.

Manuscript has been revised as follows: “These results suggest that the observed effects of PUN on cell viability and proliferation in T98G and U-87MG cells are PDIA3-mediated.”

  1. Fig. 1b: More contrast would be helpful to see the cells. Please adapt.

We tried to adjust the figure.

  1. Fig. 1a/c: Please include statistics. Check axis.

MTT assay reported in Figure 1a was done to assess the optimal concentration of punicalagin to be used in the further experiments. We did not perform statistical analysis, but it can be observed that 30 uM concentration and 48h treatment reduce the cell vitality by 50%.

In Figure 1c while untreated cells continue to proliferate up to 72h, punicalagin treatment drastically reduce cell proliferation. Statistical analysis seems unnecessary.

Figure 1c axis label has been modified in “Cell number %”

  1. Fig. 2a: Graphs are too small. Labeling of axes are not readable.

Figure 2 has been modified.

  1. For WBs: include sizes for each protein in the figure or figure legend.

Protein size in KDa has been added.

  1. Fig. 2: You directly start with (a). Heading is missing.

Heading has been added:

“Effects of PUN stimulation on PDIA3 expression and PDIA3 knockdown on cell viability and proliferation:”

  1. Fig. 4: Histograms require a higher resolution. In the main text Camptothecin is not mentioned. Why do you use that reagent? Figure legend – define CPT.

Figure 4 has been modified and the following sentence has been added to the manuscript:

 “The incubation with Camptothecin (CPT) 5 μM final concentration µM for 16 h was used as control for cell cycle arrest. “

  1. Fig. 5 Figure legend: please revise – histograms = graphs, they are not on the right.

Figure 5 has been modified.

  1. Fig. 6A: Why did you use white bands and black bands? Please unify.

Western blot analysis for H2A.X was performed using Alexa Fluor® 647 Anti-gamma H2A.X (phospho S139) antibodies. Signals were visualized as fluorescence intensity; thus, bands appear white on black background.

  1. Discussion (before last paragraph): The shown induction of apoptosis is not “pronounced”. Please revise.

Manuscript has been revised as follows:

“The results highlighted that PDIA3 inhibition increases the cell propensity to undergo apoptosis, a slightly more noticeable effect in the T98G cell line.”

Reviewer 2 Report

Autors undertaken the interesting and new subject of the possible  role of the  Protein Disulfide Isomerase A3 (PDIA3), an enzyme involved in maintaining the correct redox state of disulfide bonds in various cellular compartments , as a new target in treatment of the grade 4 glioblastoma (glioblastoma multiforme GBM). Considering  the severity of the disease as being the most comon malignant and rapidly growing tumor of CNS, the serach for a new  therapy  improving patients outcome is of great importance.  In the introduction authors given a comprehensive description  of the present knowledge considering the role of the PDIA3, that mRNA is  remarkably higher  in both low-grade gliomas (oligodendroglioma and astrocytoma) and glioblastomas, and is  linked  to poor survival in glioma patients. Investigations of the molecular mechanisms of the PDIA3 inhibition and silencing in vitro using two glioma cell lines revealed reduced glioblastoma cells spreading. Moreover, the PDIA3 inhibition increased the chemosensitivity of investigated cell lines to the approved glioblastoma drug temozolomide (TMZ).  Investigated were  many parameters of the cancer cells viability,  proliferation , cell cycle , DNA repair system. Authors used of the PUN, the previously characterized by them PDIA3 inhibitor to show its cellular cytotoxicity of both cell lines by the switch to apoptosis. The conclusion that PDIA3 might be a novel target for glioblastoma treatment , also in TMZ-resistant glioblastoma is well documented by the use of analytical methods well described with proper references. Particularly valuable is supplementary Table 1 summarizing the key findings of all cited articles. Manuscript can be published in the present form.

Author Response

Response to Reviewer 2

We are gratefully to the reviewer for the valuable comments.

Round 2

Reviewer 1 Report

The authors revised the manuscript concerning the comments. The manuscript is improved.

It´s fine.

Reviewer 2 Report

Authors performed  precise correction of the manuscript including the explanation and supplementation of all  former inconsistences. Corrected version should be accepted in  present form.